# Nomenclature and Definition of Atrophic Lesions in Small Bowel Capsule Endoscopy: A Delphi Consensus Statement of the International CApsule endoscopy REsearch (I-CARE) Group

**DOI:** 10.3390/diagnostics12071704

**Published:** 2022-07-13

**Authors:** Luca Elli, Beatrice Marinoni, Reena Sidhu, Christian Bojarski, Federica Branchi, Gian Eugenio Tontini, Stefania Chetcuti Zammit, Sherine Khater, Rami Eliakim, Emanuele Rondonotti, Jean Cristhophe Saurin, Mauro Bruno, Juliane Buchkremer, Sergio Cadoni, Flaminia Cavallaro, Xavier Dray, Pierre Ellul, Ignacio Fernandez Urien, Martin Keuchel, Uri Kopylov, Anastasios Koulaouzidis, Romain Leenhardt, Peter Baltes, Hanneke Beaumont, Clelia Marmo, Deirdre McNamara, Alessandro Mussetto, Artur Nemeth, Enrique Perez Cuadrado Robles, Guillame Perrod, Gabriel Rahmi, Maria Elena Riccioni, Alexander Robertson, Cristiano Spada, Ervin Toth, Konstantinos Triantafyllou, Gabriele Wurm Johansson, Alessandro Rimondi

**Affiliations:** 1Department of Pathophysiology and Transplantation, Università degli Studi di Milano, 20122 Milan, Italy; gianeugeniotontini@gmail.com; 2Gastroenterology and Endoscopy Unit, Fondazione IRCCS Ca’ Granda Ospedale Maggiore Policlinico, 20122 Milan, Italy; beatrice.marinoni7@gmail.com (B.M.); flaminia.cavallaro@gmail.com (F.C.); alessandro.rimondi5@gmail.com (A.R.); 3Postgraduate Specialization in Gastrointestinal Diseases, Università degli Studi di Milano, 20122 Milan, Italy; 4Department of Infection, Immunity and Cardiovascular Diseases, Royal Hallamshire Hospital, University of Sheffield, Sheffield S10 2TN, UK; reenasidhu@nhs.net; 5Charité-Universitätsmedizin Berlin, Campus Benjamin Franklin, 10117 Berlin, Germany; christian.bojarski@charite.de (C.B.); federica.branchi@charite.de (F.B.); juliane.buchkremer@charite.de (J.B.); 6Department of Medicine, Division of Gastroenterology, Mater Dei Hospital, MSD 2090 Msida, Malta; stf_che@yahoo.com (S.C.Z.); ellul.pierre@gmail.com (P.E.); 7Department of Gastroenterology, Georges-Pompidou European Hospital, 75015 Paris, France; sherine.khater@aphp.fr (S.K.); kikemurcia@gmail.com (E.P.C.R.); guillaume.perrod@aphp.fr (G.P.); gabriel.rahmi@aphp.fr (G.R.); 8Gastroenterology Department, Sheba Medical Center, Tel Aviv University, Tel Aviv 52621, Israel; abraham.eliakim@sheba.health.gov.il (R.E.); ukopylov@gmail.com (U.K.); 9Gastroenterology Department, Valduce Hospital, 22100 Como, Italy; ema.rondo@gmail.com; 10Gastroenterology Department, Hospices Civils de Lyon-Centre Hospitalier Universitaire, 69002 Lyon, France; jean-christophe.saurin@chu-lyon.fr; 11University Division of Gastroenterology, City of Health and Science University Hospital, 10126 Turin, Italy; mabru1964@gmail.com; 12Digestive Endoscopy Unit, CTO Hospital, 09016 Iglesias, Italy; cadonisergio@gmail.com; 13Centre for Digestive Endoscopy, Sorbonne University, Saint Antoine Hospital, APHP, 75012 Paris, France; xavier.dray@aphp.fr (X.D.); romain.leenhardt@aphp.fr (R.L.); 14Department of Gastroenterology, Complejo Hospitalario de Navarra, 31008 Pamplona, Spain; ifurien@yahoo.es; 15Clinic for Internal Medicine, Agaplesion Bethesda Krankenhaus Bergedorf, Academic Teaching Hospital of the University of Hamburg, 21029 Hamburg, Germany; keuchel@bkb.info (M.K.); baltes@bkb.info (P.B.); 16Department of Medicine, Odense University Hospital (OUH)-Svendborg Sygehus, 5700 Svendborg, Denmark; akoulaouzidis@hotmail.com; 17Department of Clinical Research, University of Southern Denmark (SDU), 5230 Odense, Denmark; 18Surgical Research Unit, OUH, 5000 Odense, Denmark; 19Department of Gastroenterology and Hepatology, Amsterdam University Medical Center, Location VU, 1118 Amsterdam, The Netherlands; h.beaumont@amsterdamumc.nl; 20Fondazione Policlinico Universitario Agostino Gemelli, IRCCS, Università Cattolica del Sacro Cuore, 00168 Roma, Italy; clelia.marmo@unicatt.it (C.M.); mariaelena.riccioni@unicatt.it (M.E.R.); cristianospada@gmail.com (C.S.); 21Trinity College Dublin, Tallaght University Hospital, D24 NR0A Dublin, Ireland; mcnamad@tcd.ie; 22Gastroenterology Unit, Santa Maria delle Croci Hospital, 48121 Ravenna, Italy; alessandromussetto@gmail.com; 23Skåne University Hospital Malmö, Lund University, 221 00 Lund, Sweden; artur.nemeth@med.lu.se (A.N.); ervin.toth@med.lu.se (E.T.); gabriele.wurmjohansson@skane.se (G.W.J.); 24Small Bowel Unit, Morales Meseguer Hospital, 30008 Murcia, Spain; 25Department of Gastroenterology, Western General Hospital, Edinburgh EH4 2XU, UK; alexanderrrobertson@hotmail.co.uk; 26Digestive Endoscopy and Gastroenterology Unit, Poliambulanza Foundation, 25124 Brescia, Italy; 27Hepatogastroenterology Unit, 2nd Department of Propaedeutic Internal Medicine, Medical School, Attikon University General Hospital, National and Kapodistrian University, 157 72 Athens, Greece; ktriant@med.uoa.gr

**Keywords:** small bowel atrophy, video-capsule enteroscopy, consensus

## Abstract

(1) Background: Villous atrophy is an indication for small bowel capsule endoscopy (SBCE). However, SBCE findings are not described uniformly and atrophic features are sometimes not recognized; (2) Methods: The Delphi technique was employed to reach agreement among a panel of SBCE experts. The nomenclature and definitions of SBCE lesions suggesting the presence of atrophy were decided in a core group of 10 experts. Four images of each lesion were chosen from a large SBCE database and agreement on the correspondence between the picture and the definition was evaluated using the Delphi method in a broadened group of 36 experts. All images corresponded to histologically proven mucosal atrophy; (3) Results: Four types of atrophic lesions were identified: mosaicism, scalloping, folds reduction, and granular mucosa. The core group succeeded in reaching agreement on the nomenclature and the descriptions of these items. Consensus in matching the agreed definitions for the proposed set of images was met for mosaicism (88.9% in the first round), scalloping (97.2% in the first round), and folds reduction (94.4% in the first round), but granular mucosa failed to achieve consensus (75.0% in the third round); (4) Conclusions: Consensus among SBCE experts on atrophic lesions was met for the first time. Mosaicism, scalloping, and folds reduction are the most reliable signs, while the description of granular mucosa remains uncertain.

## 1. Introduction

Capsule endoscopy (CE) has opened up a new chapter in small bowel (SB) examination by providing direct visualization of the entire SB mucosa [1].

Currently, small bowel capsule endoscopy (SBCE) and device-assisted endoscopy (DAE) are valuable tools for evaluating and characterizing SB mucosal changes. SBCE has been successfully adopted to identify SB bleedings and chronic anaemia of unknown origin. However, in the last decade, indications for its use have increased. Currently, it is employed for diagnosing and monitoring Crohn’s disease, as well as for identifying the extent and the possible malignant complications of unresponsive coeliac disease (CeD) [2,3,4,5,6]. Of note, preliminary studies suggest that SBCE can be used as a complementary technique to the histological assessment of CeD and seronegative villous atrophy (SNVA) [7].

Mucosal atrophy itself is an important endoscopic finding and, beyond its role as a hallmark of CeD, it can be identified in other inflammatory disorders of the SB. Scalloping, nodularity (also known as granular mucosa), loss of mucosal folds, and a mosaic pattern are the most commonly recognised endoscopic markers of villous atrophy [8]. These endoscopic findings suggest the presence of SB mucosal atrophy that is usually, but not always, associated with CeD [9]. However, the features of SB atrophy at SBCE are not described uniformly, and currently atrophy signs are mainly identified and verified by experts [10].

Interobserver agreement for atrophic signs is reported to vary significantly between SBCE readers. In a study by Rondonotti et al., findings from 32 patients with CeD and 11 patients with normal duodenal mucosal biopsies were examined by 4 reviewers. Kappa values for interobserver agreement ranged between 0.56 and 0.87 [11]. In contrast, Biagi et al. found higher interobserver variability, with kappa values of 0.49, 0.67, and 0.70 between 3 reviewers reading 32 SBCEs with 26 confirmed CeD, suggesting that experience is needed in interpreting SBCEs [12]. In a study by Petroniene et al. that examined SBCEs for 10 CeD patients and 10 controls, interobserver agreement was excellent (κ = 1.00) when only experienced reviewers were considered, whereas agreement decreased when reviewers with little prior experience were considered (κ = 0.20) [13]. Interobserver agreement may also vary according to different CeD characteristics. For example, Murray et al. showed that overall interobserver agreement was highest (κ = 0.77) for the mosaic pattern in 38 patients with CeD who underwent SBCE, followed by scalloping (κ = 0.59), fissuring (κ = 0.41), and villous atrophy (κ = 0.37) [14].

Therefore, as suggested by Jang et al., expertise in reading SBCE with the adoption of structured terminology for capsule findings would likely demonstrate an improvement in reader agreement [15,16].

The aim of our study was to reach international consensus among experts regarding the nomenclature and the description of the main atrophic lesions in SBCE. This effort is part of a wider initiative of the International Capsule endoscopy REsearch (I CARE) Group to reach agreement on the nomenclature and the descriptions of CE findings with a Delphi method-based approach [17,18,19].

## 2. Materials and Methods

### 2.1. Centres and Design

We adopted the Delphi technique to reach an agreement among a panel of European gastroenterologists working in tertiary referral centres for CE. Particular attention was paid to the selection of CeD specialists with substantial publications on SBCE and CeD, given their particular expertise in atrophic mucosal lesions. Additionally, for each participant, we recorded expertise in CE (e.g., years of activity, number of capsules read per year) as well as their specific expertise in reading CE from coeliac patients.

We employed web-based questionnaires (Google LLC) to collect the responses of the expert panel. In addition, we employed a six-point Likert scale (i.e., strongly disagree, disagree, moderately disagree, moderately agree, agree, strongly agree) to quantify the agreement of the experts on nomenclature, description, and the matching of proposed images of atrophic lesions frequently found on SBCE in coeliac patients.

### 2.2. Delphi Process and Phases

In the first phase, taking into consideration the relatively low prevalence of SBCE in coeliac patients, we decided to submit the nomenclature and the descriptions of atrophic lesions to a core group of 10 experts with substantial expertise in CeD, with the aim of reaching widespread and shared agreement on these definitions (CB, FB, SCZ, SK, LE, RE, ER, JCS, RS, GET). We set a higher cut-off threshold of 90% of total ‘strongly agree’ and ‘agree’ responses to reach consensus in this core group. This precaution was deemed necessary to strengthen the acquisitions of this first Delphi round. For each Delphi question, we collected the core group comments and suggestions on nomenclature and definitions that were later considered in the following Delphi rounds if the proposed agreement was not reached.

After consensus on nomenclature and description was reached in the core group, we collected a set of images of atrophic lesions from a large database of CE photographs (>1200) of histologically proven atrophic SB mucosa (i.e., Marsh 3 lesions) [20,21]. We submitted the selected images to a group of 36 endoscopists (including the 10 experts of the core group). For each group of images, the experts had to decide whether the images fit the agreed descriptions and nomenclature of the atrophic lesion. In addition, the experts had to decide if any images did not perfectly match the description or if none matched. In this second part of the Delphi process, we set a priori a threshold of 80% of the total ‘strongly agree’ and ‘agree’ responses to meet the definition of consensus. When consensus was not met, we checked if the opinions of the experts regarding replacement of any picture were statistically significant (Figure 1).

### 2.3. Statistical Analyses

We employed chi-squared analysis to statistically examine our results, considering a random 50:50 probability of choosing between a positive (i.e., one or some images should not be considered) and negative (i.e., all the images are suitable) response. If a majority of the endoscopists gave a positive response, we would consider a 25% probability that a single selected picture would be defined by chance as not suitable and then looked for any picture that statistically emerged as not suitable. The author in charge of the selection of the images (LE) took these results into account and accordingly changed the proposed set of images in the following Delphi round.

Any statements or any set of pictures that did not meet consensus after three Delphi rounds was considered a failure.

We employed R Studio 4.0.0 (R Core Team (2020). R: A language and environment for statistical computing. R Foundation for Statistical Computing, Vienna, Austria, https://www.R-project.org/) for statistical analyses.

The study was conducted in compliance with the Helsinki criteria and all subsequent amendments. The data were collected within the framework of standard patient care. They were treated confidentially, in compliance with the most recent privacy laws at the European and the national level, and anonymized so that those who analysed the data were not able to identify patients (protocol number 137/2021).

## 3. Results

We recruited 36 SBCE experts from 11 countries (i.e., France, Germany, Greece, Ireland, Israel, Italy, Netherlands, Poland, Spain, Sweden, United Kingdom). Our panel of experts had a median of 13 (IQR 10–19) years’ experience in CE, a mean of 100 (IQR 50–150) CE exams annually and a mean of 5 (IQR 2–11) capsule exams of coeliac patients annually.

The core group identified four main atrophic lesions that are frequently found in SBCE: mosaicism, scalloping, folds reduction, and granular mucosa.

The consensus in the core group (*n* = 10) on the definitions and the nomenclature of these findings was met in the second round for mosaicism, scalloping, and folds reduction, with an overall agreement of 90%, 100%, and 100%, respectively, and in the third round for granular mucosa, with an overall agreement of 90% (Table 1 and Table 2)**.**

Afterward, we collected the responses of the larger panel of experts on the agreement between the established nomenclature and the definitions and the proposed selection of images. The consensus was met in the first round for mosaicism (88.9%), scalloping (97.2%), and fold reduction (94.4%) (Table 3, Figure 2A–C).

Granular mucosa did not reach consensus agreement after three rounds and therefore the Delphi process was halted. In the final round, a global consensus of 75% on the proposed set of images was reached (Figure 2D)**.** In the analysis of the results for this round, we found that the experts pointed out that the left upper quadrant image did not match the given description of granular mucosa (*p* = 0.001). In post hoc analysis, we looked for any statistically significant association between agreement on the proposed set of images of granular mucosa and overall experience in SBCE, the number of SBCEs per year, and the number of dedicated coeliac SBCEs per year. In particular, we aimed at finding any differences in judgment among the endoscopists in the upper quartile of these categories and the rest of the endoscopists. Of note, we found that years’ experience in SBCE and expertise in reading SBCEs from coeliac patients did not statistically influence decisions on agreement regarding granular mucosa (*p* = 0.21 and *p* = 0.28, respectively). On the contrary, the endoscopists in the upper quartile for numbers of SBCEs read per year were more prone to disagreement than the others, regarding the final set of proposed images for granular mucosa (*p* = 0.03).

## 4. Discussion

For the first time, this study has established agreement among 36 experts on the definitions of the most common atrophic lesions found in SBCE through a Delphi consensus. This initiative is part of the I CARE Group effort to improve interobserver agreement on CE through the adoption of shared and agreed nomenclature and definitions of CE findings.

We have found agreement on the terminology (i.e., nomenclature and description) of all the atrophic items commonly reported in the literature: mosaicism, scalloping, folds reduction and granular mucosa [8]. Agreement of endoscopic appearance, based on a large database of histologically proven atrophy (i.e., Marsh–Oberhuber 3) photographs of SBCEs, was met for scalloping, mosaicism, and folds reduction. Granular mucosa was the only item that lacked an agreed endoscopic appearance.

Although there is an existing SBCE terminology [16,17,18,19,22], there is still a lack of consensus among experts on the definition and the nomenclature of the most common features of atrophy. With the increasing use of video CE in coeliac patients, especially those with refractory disease, the need for standardization of SBCE reading, reporting, nomenclature and definitions, and the typical appearance of atrophic small bowel lesions has become apparent [23]. This particular need is going to become even more important in the next few years, as the use of artificial intelligence (AI) software for endoscopy diagnosis is expected to markedly increase. Large CE databases with solid and shared findings between experts are keystones for effective machine learning and for building precise and trustworthy AI programs [24,25]. The results of our work have shown that the main features of atrophic lesions are mosaicism, scalloping, and folds reduction, and they have rapidly found consensus among readers of SBCEs, even among those who read only a few CEs of coeliac patients per year. This finding was relatively expected since they are the most recognizable hallmarks of atrophy [26]. In our study, scalloping showed the highest level of agreement for nomenclature, definition, and endoscopic appearance. This is in line with the results of other studies that suggested a strong specificity (99%) and a positive predictive value (97%) for villous atrophy [27]. For mosaicism and folds reduction, the current literature is less clear. However, they demonstrated a high diagnostic accuracy for CeD (98.1%) when associated with scalloping [26]. In our experience, folds reduction shows a slightly higher rate of agreement for description and endoscopic appearance than mosaicism. Notably, during the Delphi process, some doubts were raised about the possibility of finding only a mosaicism pattern without scalloping.

Conversely, the finding ‘granular mucosa’ achieved agreement on its nomenclature and description in the core group of 10 experts but failed to achieve validation in the expert group regarding its assignment to 4 typical images. As mentioned earlier, we found no statistically significant association between agreement on the proposed images and expertise in SBCE in CeD or years’ experience in SBCE. A theoretical general lack of specificity and the presence of similar findings in other small bowel diseases (e.g., oedema in inflammatory bowel disease) may be the main factors that prevented general agreement on this finding [17]. Notably, in the final Delphi discussion for ‘granular mucosa’, the panel of experts presented some concerns about the proposed images. They stated that mucosal changes could be easily mistaken with inflammatory changes or not specific mucosal alterations. Moreover, this finding shows that expertise in coeliac SBCE is not the key factor for identifying the most common atrophic lesions, hinting at the possible ease of reproducibility of our work. With the statistically proven lower agreement in the upper quartile of endoscopists by numbers of SBCEs, it becomes clear that the main features of atrophy may be identifiable by experts in SBCE but not necessarily trained in coeliac SBCE.

There are some limitations to this study. The selection of images from one single database may represent potential subjective bias. However, on the other hand, this particular feature of the study may be balanced by the verified histological confirmation of atrophy for each image, which was usually obtained in a specific tertiary referral center. The web-based process could have a limited discussion and debate on the images. Moreover, the consensus aimed at providing nomenclature for and definitions of hallmarks of mucosal atrophy rather than esteeming their clinical relevance in terms of severity of atrophy. Furthermore, a potential limitation in a real-life scenario is that more than a finding can be present in the same still frame, hindering the possible reproducibility of our work in the hands of non-expert readers. However, the proposed and accepted images could support non-expert readers to recognize single hallmarks of atrophy at SBCE.

In conclusion, our international group of SBCE experts has reached consensus on the nomenclature, description, and identification of atrophic lesions in SBCE. The current study is an important contribution to achieving higher quality SBCE reading and reporting; and, it provides a common technique for evaluating coeliac patients—particularly those with refractory disease—although it must be underlined that atrophy does not always mean CeD [9]. We strongly suggest the adoption of mosaicism, scalloping, and folds reduction when describing atrophy in SBCE. The description of granular mucosa remains unspecific, and it should only be used with caution in centers with a very high volume of SBCEs.

## Figures and Tables

**Figure 1 diagnostics-12-01704-f001:**
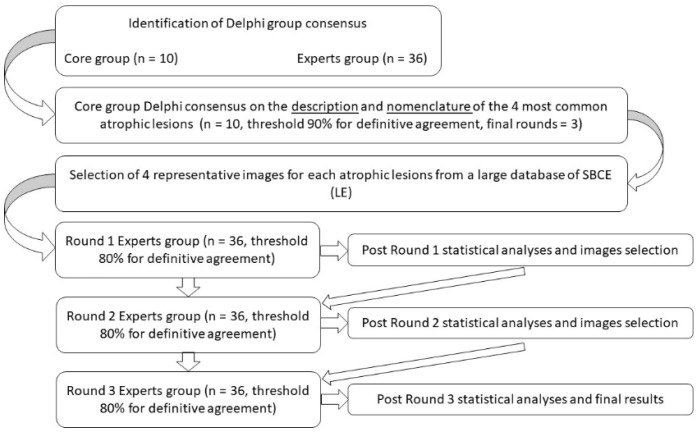
Flowchart of the Delphi process.

**Figure 2 diagnostics-12-01704-f002:**
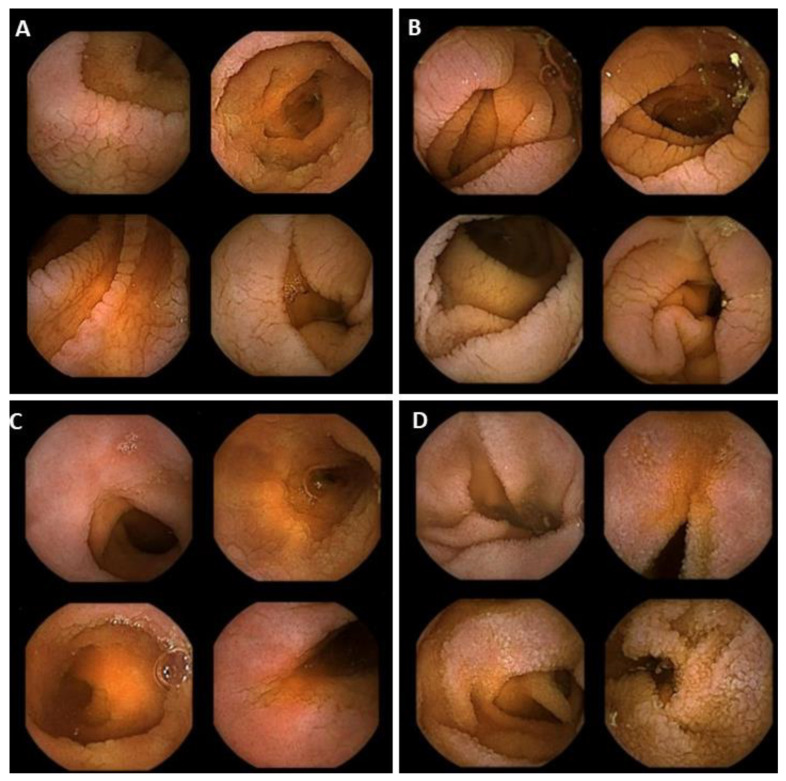
Agreed images representing the endoscopic findings suggestive for villous atrophy: mosaicism (**A**), scalloping (**B**), folds reduction (**C**), and granular mucosa (**D**).

**Table 1 diagnostics-12-01704-t001:** Core group agreement for nomenclature and description of common atrophic lesions.

Nomenclature and Description	Agreement in Core Group (*n* = 10)	Number of Rounds
Mosaicism: loss of villous structure with the presence of non-ulcerated, orthogonally converging fissures of the small bowel mucosa	9 out of 10 (90%)	2
Scalloping: presence of multiple incisures on the edge of the small bowel folds (cogwheel appearance)	10 out of 10 (100%)	2
Folds reduction: flattening of the mucosa with reduction of the folds (<2 field view) in terms of both height and number	10 out of 10 (100%)	2
Granular mucosa: mucosal surface characterized by multiple small nodules, rough villous architecture and edema of the villi	9 out of 10 (90%)	3

**Table 2 diagnostics-12-01704-t002:** Core group rating on the nomenclature and description of common atrophic lesions.

Nomenclature and Description	Numerical Scale/Expert Votes						% Agreeing and Strongly Agreeing	Number of Rounds
	Strongly Disagree	Disagree	Moderately Disagree	Moderately Agree	Agree	Strongly Agree		
Mosaicism	0	0	0	1	5	4	90	2
Scalloping	0	0	0	0	4	6	100	2
Folds Reduction	0	0	0	0	2	8	100	2
Granular Mucosa	0	0	0	1	4	5	90	3

**Table 3 diagnostics-12-01704-t003:** Experts group rating on the correspondence between suggested images and core group definition. Consensus was not reached after three rounds for ‘Granular Mucosa’.

Images	Numerical Scale/Expert Votes						% Agreeing and Strongly Agreeing	Number of Rounds
	Strongly Disagree	Disagree	Moderately Disagree	Moderately Agree	Agree	Strongly Agree		
Mosaicism	0	1	1	2	16	16	88.9	1
Scalloping	0	0	1	0	8	27	97.2	1
Folds Reduction	0	1	0	1	18	16	94.4	1
Granular Mucosa	0	3	1	5	20	7	75.0	3

## Data Availability

Data are available by correspondence to the Corresponding Author.

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
