# Peer review of "Nomenclature and Definition of Atrophic Lesions in Small Bowel Capsule Endoscopy: A Delphi Consensus Statement of the International CApsule endoscopy REsearch (I-CARE) Group"

_diagnostics, 2022, doi:10.3390/diagnostics12071704_

Round 1
Reviewer 1 Report
The manuscript is generally well-written and structured and a standardization of the nomenclature is definitely a major need among gastroenterology specialists. Nevertheless, I would suggest the authors to improve the section of discussions and to add more details regarding each mucosal change and perhaps even mention the pros and cons regarding the agreement on nomenclature.
Author Response
We thank the reviewer for the meaningful suggestions.
Accordingly, we improved the introduction and discussion sections. We added new references (26-27) and a new paragraph (page 8, line 12) to discuss the pro and cons of the consensus for non-expert SBCE capsule readers. We re-reviewed the manuscript and improved some minor English mistakes.
Please find attached the revised manuscript.
Reviewer 2 Report
nothing to write but nice work !
Author Response
We thank the reviewer
Round 2
Reviewer 1 Report
I appreciate that you followed the recommendations, but the suggestion was to describe each mucosal change more throroughly and to mention the reasons for which the experts disagreed regarding the nomenclature.
Author Response
We thank the referee for the suggestion
Accordingly, we have discussed more in details the different endoscopic signs and the reason of disagreement about the granular mucosa (pages 7-8). This sign was basically considered unspecific and frequently correlated to inflammation
Round 3
Reviewer 1 Report
Thank you for your explanations.